# Deep Neural Networks and Machine Learning Radiomics Modelling for Prediction of Relapse in Mantle Cell Lymphoma

**DOI:** 10.3390/cancers14082008

**Published:** 2022-04-15

**Authors:** Catharina Silvia Lisson, Christoph Gerhard Lisson, Marc Fabian Mezger, Daniel Wolf, Stefan Andreas Schmidt, Wolfgang M. Thaiss, Eugen Tausch, Ambros J. Beer, Stephan Stilgenbauer, Meinrad Beer, Michael Goetz

**Affiliations:** 1Department of Diagnostic and Interventional Radiology, University Hospital of Ulm, Albert-Einstein-Allee 23, 89081 Ulm, Germany; christoph.lisson@uniklinik-ulm.de (C.G.L.); marc.mezger@uni-ulm.de (M.F.M.); daniel.wolf@uniklinik-ulm.de (D.W.); stefan.schmidt@uniklinik-ulm.de (S.A.S.); wolfgang.thaiss@uniklinik-ulm.de (W.M.T.); meinrad.beer@uniklinik-ulm.de (M.B.); michael.goetz@uni-ulm.de (M.G.); 2Center for Personalized Medicine (ZPM), University Hospital of Ulm, Albert-Einstein-Allee 23, 89081 Ulm, Germany; ambros.beer@uniklinik-ulm.de; 3Artificial Intelligence in Experimental Radiology (XAIRAD); 4Visual Computing Group, Institute of Media Informatics, Ulm University, James-Franck-Ring, 89081 Ulm, Germany; 5Department of Nuclear Medicine, University Hospital of Ulm, Albert-Einstein-Allee 23, 89081 Ulm, Germany; 6Department of Internal Medicine III, University Hospital of Ulm, Albert-Einstein-Allee 23, 89081 Ulm, Germany; eugen.tausch@uniklinik-ulm.de (E.T.); stephan.stilgenbauer@uniklinik-ulm.de (S.S.); 7Comprehensive Cancer Center Ulm (CCCU), University Hospital of Ulm, Albert-Einstein-Allee 23, 89081 Ulm, Germany; 8Center for Translational Imaging “From Molecule to Man” (MoMan), Department of Internal Medicine II, University Hospital of Ulm, Albert-Einstein-Allee 23, 89081 Ulm, Germany; 9i2SouI—Innovative Imaging in Surgical Oncology Ulm, University Hospital of Ulm, Albert-Einstein-Allee 23, 89081 Ulm, Germany; 10German Cancer Research Center (DKFZ), Division Medical Image Computing, Im Neuenheimer Feld 280, 69120 Heidelberg, Germany

**Keywords:** radiomics, machine learning, deep learning, deep neural networks, personalised oncology, precision imaging

## Abstract

**Simple Summary:**

Mantle cell lymphoma (MCL) is an aggressive lymphoid tumour with a poor prognosis. There exist no routine biomarkers for the early prediction of relapse. Our study compared the potential of radiomics-based machine learning and 3D deep learning models as non-invasive biomarkers to risk-stratify MCL patients, thus promoting precision imaging in clinical oncology.

**Abstract:**

Mantle cell lymphoma (MCL) is a rare lymphoid malignancy with a poor prognosis characterised by frequent relapse and short durations of treatment response. Most patients present with aggressive disease, but there exist indolent subtypes without the need for immediate intervention. The very heterogeneous behaviour of MCL is genetically characterised by the translocation t(11;14)(q13;q32), leading to Cyclin D1 overexpression with distinct clinical and biological characteristics and outcomes. There is still an unfulfilled need for precise MCL prognostication in real-time. Machine learning and deep learning neural networks are rapidly advancing technologies with promising results in numerous fields of application. This study develops and compares the performance of deep learning (DL) algorithms and radiomics-based machine learning (ML) models to predict MCL relapse on baseline CT scans. Five classification algorithms were used, including three deep learning models (3D SEResNet50, 3D DenseNet, and an optimised 3D CNN) and two machine learning models based on K-nearest Neighbor (KNN) and Random Forest (RF). The best performing method, our optimised 3D CNN, predicted MCL relapse with a 70% accuracy, better than the 3D SEResNet50 (62%) and the 3D DenseNet (59%). The second-best performing method was the KNN-based machine learning model (64%) after principal component analysis for improved accuracy. Our optimised CNN developed by ourselves correctly predicted MCL relapse in 70% of the patients on baseline CT imaging. Once prospectively tested in clinical trials with a larger sample size, our proposed 3D deep learning model could facilitate clinical management by precision imaging in MCL.

## 1. Introduction

Mantle cell lymphoma (MCL) is a rare subtype of lymphoid malignancy, representing 2.5–6% of all non-Hodgkin lymphoma (NHL) [1,2]. MCL is considered a heterogeneous disease with various clinical presentations ranging from minimal symptoms to progressive generalised lymphadenopathy [3,4,5,6]. Most MCL patients present with aggressive disease in an advanced stage (Stage III-IV); however, a subset of patients exhibit an indolent evolution [7]. As a result, the World Health Organization (WHO) updated the classification of MCL in 2017, describing two main subtypes, “classical MCL” and “indolent leukemic non-nodal/smouldering MCL”, with significant differences in the molecular characteristics, clinical features and prognosis [8]. According to follow-up studies, early identification of high-risk patients may modify their therapeutic management, reduce unnecessary toxicity, and improve prognosis [9,10,11,12].

As initial staging, the ESMO guidelines recommend performing a contrast-enhanced computed tomography (CT) scan of the neck, thorax, abdomen and pelvis as treatment differs depending on the stage of the disease [7]. A PET/CT scan is especially recommended in the rare limited stages I/II before localised radiotherapy, although reimbursement by the domestic health insurances is not ensured. There is consensus to base diagnosis upon histopathological evaluation of a lymph node biopsy rather than a fine needle aspiration, keeping in mind the heterogeneity of MCL [7].

Novel therapeutic options are constantly evolving, but until today, management of MCL is challenging due to its pattern of resistance, early relapse and poor long-term survival [13]. Despite improvements in remission durations, MCL is regarded as an incurable malignancy, despite the novel drug strategies developed in recent years [14,15].

The “mantle cell lymphoma International Prognostic Index (MIPI)”, based on age, ECOG performance status, lactate dehydrogenase and leukocyte count, assigns patients to low-risk, intermediate-risk and high-risk groups with 5-year survival rates of 60%, median overall survival of 51 months or median overall survival OS 29 months, respectively [16]. Furthermore, the cell proliferation antigen Ki-67 is considered the most established histomorphological risk factor in MCL [2]. Combining the Ki-67 index with the MIPI led to the more refined MIPI-c [17]. Additional TP53 gene mutation at diagnosis is associated with poor response to upfront intensive chemotherapy and poor prognosis [4].

The prognostic potential of minimal residual disease (MRD) has been confirmed in numerous studies [18,19]. Due to current limitations, for example, how to react in MRD-positive patients, MRD is tested in clinical trials but not recommended in clinical routine. Currently, no biomarkers definitely predict indolent behaviour; instead, a short ‘watch and wait’ period under close observation seems appropriate in indolent cases with low tumour burden [14,20].

The treatment selection of any given MCL patient is based mainly on clinical and biological risk factors, symptoms and tumour load [7]. In the era of precision oncology, there is great enthusiasm to identify tumour-specific targetable alterations for precise management for the individual patient, maximising the success of treatment by minimising the therapy-related toxicities.

Most research in precision medicine focuses on molecular tumour profiling using genomics-based arrays from biopsies with promising results in clinical oncology [11], including haematological diseases [21]. However, tumours are spatially and temporally heterogeneous, i.e., demanding repeated biopsies to capture their molecular heterogeneity with an increased risk for the patient [22,23,24,25].

Advanced medical imaging addresses the challenges of biopsy being non-invasive, repeatable and applicable for difficult-to-reach lesions within the body. The emergence of high-resolution image acquisition and modern computational technologies amid the requirements of precision medicine have given rise to artificial intelligence (AI)—based image analysis in radiology. In general, AI techniques detect specific patterns in medical images using data characterisation algorithms to convert conventional imaging information into quantitative mineable “big data” [11,26]. Radiomics uses imaging-based texture analysis to generate imaging features extracted from a region of interest (ROI) to reflect tumour physiology and radiographic phenotype for classification, prognosis and monitoring of treatment response in many cancer types, including lymphoma [12,26,27,28,29,30,31,32,33,34,35,36,37,38,39,40,41,42,43,44,45,46,47,48,49,50,51,52,53]. Additionally, there are indications of an association between radiomic features and the underlying gene expression patterns [54]. Combining radiomic image analysis with classic machine learning algorithms is an emerging technique. Machine-learning (ML) refers to algorithms that analyse and learn on a dataset before making decisions or predictions based on the learned information [55,56]. Radiomics-based ML has been shown to predict primary treatment failure in diffuse large B-cell lymphoma (DLBCL) better than standard clinical criteria [45]. However, high-throughput radiomic analysis is limited by manual delineation of tumour boundaries, which is labour intensive with poor reproducibility.

Deep learning (DL) algorithms are considered advanced AI techniques and have emerged as a promising tool in many research fields [57,58]. DL is based on complex artificial neural networks that detect patterns in the data and learn efficient features from the raw data without the preprocessing steps required in classic machine learning models [59]. For image analysis, deep neural networks based on convolutional neural networks (CNNs) have proven efficient in accurately locating and segmenting tumour lesions, useful for diagnosis and image analysis tasks, and outperforming other traditional automated algorithms [60].

To the best of our knowledge, this is the first study to develop and compare 3D deep neural networks with radiomics-based machine learning methods on routine CT imaging at baseline for predicting relapse of mantle cell lymphoma (MCL), thus promoting precision imaging in clinical oncology.

## 2. Materials and Methods

### 2.1. Patients and Imaging Protocol

Enrolled in this retrospective study were 30 treatment-naive patients with histologically proven mantle cell lymphoma who underwent contrast-enhanced CT or PET/CT scans at our institution between January 2005 and January 2016. Demographic patient data, laboratory and clinical data (such as the white blood cell count, lactate dehydrogenase levels and Ki-67 proliferation), disease stage according to the Ann Arbor staging system, treatment regimen and clinical outcome data were recorded by thoroughly reviewing electronic charts and the radiology information system.

Incomplete clinical and imaging data records and lack of histological confirmation were exclusion criteria. Further exclusion criteria included patients whose disease status was not confirmed at the end of therapy and those who had disease progression within the first line of treatment.

Additional inclusion criteria were treatment with an R-CHOP-based regimen (rituximab, cyclophosphamide, doxorubicin, vincristine and prednisone) alternating with or instead of R-DHAP (rituximab, dexamethasone, high-dose cytarabine and cisplatin) or R-FC (rituximab, fludarabine and cyclophosphamide).

The primary endpoint was relapse of MCL based on clinical assessment and imaging studies. Follow-up recordings in the electronic charts determined each patient’s status.

All enrolled MCL subjects were followed up for at least five years after therapy, except those who experienced death by any cause. A flow diagram of the cohort selection is presented in Figure 1.

Contrast-enhanced PET/CT or CT images for assessment of disease status were performed close to therapy initiation. The images used for analysis were acquired on the multiple-row detector CT scanners Philips Brilliance CT 64-channel scanner or Philips Brilliance iCT 256-channel scanner (Philips Healthcare, Cleveland, OH, USA) and the PET/CT scanner Siemens Biograph mCT(40)S (Siemens Healthineers, Erlangen, Germany). Because most MCL patients in our retrospective study had been enrolled in clinical trials, such as the MCL R2 Elderly trial (NCT01865110), or had been treated according to them, all CT and PET/CT imaging used the acquisition and reconstruction parameters according to the standard protocol:

CT scans were performed after intravenous contrast agent injection of Ultravist^®^ 370 (Bayer Schering Pharma, Berlin, Germany) in a weight-adopted dose with a delay of 70–80 s to represent the portovenous phase of chest and abdomen (tube voltage 100–120 kV with automatically calculated tube current; collimation 40 × 0.625 mm and 64 × 0.625 mm; 1 mm reconstruction thickness, increment 0.5 mm, matrix 512 × 512). Thin-slice CT images were reconstructed using a soft tissue kernel (filter, B31f) for further analysis.

18F-FDG-PET/CT scans were performed approximately 60 min after intravenous administration of 18F-FDG with a mean of 282 MBq of (range 223–334 MBq). The contrast-enhanced CT image was acquired in the portal venous phase 70–80 s after intravenous contrast agent injection (Ultravist 370, Bayer Schering Pharma, Berlin, Germany) in a weight-adopted dose. Attenuation-based online modulation of tube current (CARE Dose) was used with a quality reference tube current setting (reference mAs) of 210 mAs (tube voltage 120 kV; 16 × 1.2-mm collimation; 4 mm slice thickness; matrix 512 × 512) followed by the PET scan from the mid-thighs to the base of the skull in 5 to 8 bed positions.

All PET scans were acquired in 3D mode with an acquisition time of 3 min per bed position in the time of flight technique. PET data were reconstructed with attenuation correction using dedicated standard software (PETsyngo, Siemens, Erlangen, Germany).

### 2.2. Radiomic-Based Machine Learning as Prediction Model

#### 2.2.1. Tumour Segmentation, Data Processing and Feature Extraction

Two board-certified radiologists analysed all images, both with over 10 years of experience in oncologic imaging and over 5-year experience in texture analysis. Consent was established by joint consultation in cases of disagreement. Target lymphoma lesions were selected based on Cheson criteria [40]. Segmentation, texture analysis and feature extraction were performed using the software mint Lesion™ (mint Medical GmbH, Heidelberg, Germany), which allows the three-dimensional size and whole lesion radiomic measurements.

All target lymph nodes were segmented in a semiautomatic process by manually delineating superior and inferior tumour boundaries. The software algorithm computed the tumour boundaries on the slices in between for “whole tumour volume” data. The final 3D segmentation was thoroughly reviewed, and if necessary, the tumour boundaries were manually modified. In the final processing step, radiomic features were quantified concerning their characteristic pattern of grey levels within the ROI using texture feature descriptors according to the Image Biomarker Standardization Initiative (IBSI) guidelines [38]. There were 22 texture features of first-order and 26 features of second-order derived from the grey-level co-occurrence matrix were chosen for further analysis (see Table A2 in Appendix A). First-order statistics are histogram-based methods that describe the distribution of values of individual voxels and reduce a region of interest to single values for mean, median, maximum, minimum, uniformity and others. Second-order statistics consider spatial relationships as they describe statistical interrelationships between voxels with similar or dissimilar contrast values. Image-based texture analysis was first introduced in 1973 by Haralick et al. [61].

The 3D volumetric radiomic features were extracted and used as input volume for machine learning model development. See Figure 2 for a depiction of the workflow.

#### 2.2.2. Features Selection

After preprocessing, the number of features was reduced using a filtering-based feature selection. Like any other data-mining application, radiomics underlies the curse of dimensionality [62], and an appropriate feature selection strategy is crucial to reduce the great number of initially extracted quantitative radiomic features. By choosing a subset of optimal features, overfitting is reduced, and thus models experience increased generalizability for simpler and faster ML models with enhanced performance [63]. After preprocessing, feature selection using filter feature selection techniques was applied [63]. Like any other data-mining application, radiomics underlies the curse of dimensionality [62], as it extracts a large number of quantitative features from the regions of interest (ROIs).

An appropriate feature selection strategy is crucial to reduce the dimensionality of radiomic features; by choosing a subset of optimal features, overfitting is reduced, and thus models experience increased generalizability for simpler and faster ML models with enhanced performance [63].

For this study, RelieFF—a well-known and often utilised feature selection method—was used. The top 50 features were chosen as the input variable for further analysis, as shown in the correlation plot in Figure 3. The calculations were done using the open-source software licensed under the GNU General Public License Weka Toolkit (version 3.8, University of Waikato, Waikato, New Zealand) [64,65,66,67,68,69,70].

#### 2.2.3. Dataset Characteristics and Preprocessing

The dataset consisted of 134 ROIs in the CT examinations of the 30 MCL patients (17 patients with complete remission and 13 patients with relapse of disease). Data were randomly split into training (70%) and testing sets (30%) but stratified to ensure consistent distribution in the training and testing sets to mimic the real-world distribution and reduce overfitting. The splitting was done considering the patient level, i.e., no patient contributed to training and testing.

The selected radiomic features of the 134 ROIs were the input variables. The output variables were binary: The patients who suffered from a relapse of disease within the 5-year observation period were assigned to the high-risk group, whereas those with complete remission were classified in the low-risk group. The selected features were used for the supervised machine learning model, including RF and KNN.

#### 2.2.4. Machine Learning (ML) Classification Architectures

Our machine learning pipeline used two classic machine learning algorithms—Random Forest (RF) and K-nearest Neighbor (KNN) architectures. The medical imaging community has proven these methods efficient and suitable [71,72,73]. The algorithms were implemented using the open-source Python machine learning library Scikit-learn [74].

The Random Forest algorithm (RF) [75,76] is an ensemble classifier that produces multiple decision trees using randomly selected subsamples of the data set. The prediction is achieved by averaging the predictions of all decision trees. This improves the shortcomings of individual decision tree algorithms—namely, weak generalisation and a tendency to overfitting. We used the default RandomForestClassifier from sklearn.

The K-Nearest Neighbors algorithm (KNN) [77] is also known as “Instance-based” learning as the hypotheses are built from the training instances. KNN is based on distance calculations between instances: Target data points are classified according to the labels of the closest neighbours from the training data. KNN is a conceptually simple but powerful algorithm that is easy to use, interpret and implement. We used the grid search optimisation algorithm to fine-tune the model and selected five neighbours in our study.

#### 2.2.5. ML Model Development and Evaluation

RF and KNN algorithms were trained and selected by comparing performance on the test set’s evaluation. Each algorithm used ten seeds for the training and testing split to ensure inter-model reproducibility. Results of all runs were averaged and standard deviation and confidence intervals were calculated.

All computing was done on a single workstation running Ubuntu 20.04 LTS with 32 GB memory, AMD Ryzen 7 3700X (AMD, Sunnyvale, CA, USA) and NVidia RTX 2070 Super 8 GB VRAM (Santa Clara, CA, USA). The following software modules were installed: Scikit-learn 0.23.3, Python 3.8.5, JupyterLab 2.2.6. (Further details see below).

### 2.3. Neural Network Approach as Prediction Model

Convolutional neural networks (CNNs) are deep learning techniques useful for diagnosis and image analysis tasks [78,79]. In general, a CNN consists of a series of layers of convolution filters followed by fully connected layers at the very end of the network. In contrast to traditional ML models, CNNs do not require the quantification and selection of radiomic features as they are trained directly on the images.

Three different network architectures were trained and evaluated—an optimised 3D CNN designed by ourselves and two larger state-of-the-art 3D CNNs, consisting of a 3D DenseNet and a 3D Squeeze-and-Excitation ResNet 50 (3D SE ResNet). Details of the three networks are described in the “network architecture” section. See Figure 4 for a depiction of the workflow.

#### 2.3.1. Dataset, Training and Validation

The dataset consisted of CT examinations containing 134 ROIs of 30 MCL patients. ROIs were dichotomised into MCL in complete remission (CR) and MCL with relapse of disease (RD); 35% were randomly selected for testing, while the remaining 65% were used for training and validation. The training set was utilised for training the network, while the validation set was used to monitor the effectiveness of the training. ROIs of one patient were either in the training, validation or testing group, but never in more than one at a time.

At first, 3D cubes of each target lymphoma lesion with the tumour’s centre of mass at the centre of the cube were cropped from CT images. After evaluating the range of target sizes, we determined one cube measuring 20 × 80 × 70 mm to fit each target lesion completely within the cube without overlapping a potential neighbouring target lesion.

The original dataset was preprocessed using the above-described methods to generate 3D patches of 20 × 80 × 70 pixels focusing only on the ROI. The adaptive moment estimation (Adam) optimiser was used to minimise the cross-entropy loss between the model outputs and target classification labels. Adam optimiser was configured with a learning rate of 0.001 and a dropout of 30% to prevent overfitting of our model. The loss of the output class “relapse of disease” was double weighted. Due to the limited amount of training data in our data set, we used augmentation techniques for image processing “Random Flip”, “Gaussian Blur” and “Gaussian Noise” with Torchvtk (https://github.com/torchvtk/torchvtk; last accessed 15 January 2022) to enhance the robustness and to prevent overfitting [78].

Additionally, this ensures the model focuses on the target lesions rather than various sources of noise [80]. We stopped the training after 100 epochs.

The duration of training was 2 h per run, with a batch size of 4 determined by GPU memory. After every training epoch, the network was evaluated with the validation dataset and the model from the epoch with the best validation score was used for classifying the test set. We tested the classification performance of all three network architectures with five runs with five different seeds for the training and validation dataset.

We evaluated the DL models by accuracy, precision, recall, F1-Score and the area under the curve (AUC) by calculating the mean and standard deviation over the five runs. To assess the clinical value of the predictive models, we performed a decision curve analysis.

The following software modules were installed: PyTorch 1.9, Monai 0.6, Python 3.9.5, JupyterLab 3.0.16, NumPy 1.21.0. All code, from data preprocessing to model building, was written in Python 3.9.5 based on the open-source deep-learning library “PyTorch”, together with the open-source framework for deep learning in medical imaging “Monai” (https://monai.io/; last accessed 15 January 2022).

Neural Networks were trained with CT images on a single workstation running Ubuntu 20.04 LTS with 32 GB memory, AMD Ryzen 9 3900X (AMD, Sunnyvale, CA, USA) and NVidia RTX 3080 10 GB VRAM (Santa Clara, CA, USA).

#### 2.3.2. Deep CNN Architecture

Following are the architectural design details of the model used:

3D DenseNet: We used the 3D DenseNet-121 [81] from Monai [Medical Open Network for Artificial Intelligence MONAI https://monai.io/; last accessed 15 January 2022]. Densely connected convolutional networks (short: DenseNets) consist of several dense blocks. Each layer in one dense block is connected to all other layers with feed-forward connections, thus requiring fewer parameters than traditional CNNs. Advantages are reduced overfitting for smaller training sets and improved vanishing gradient problems.

3D Squeeze-and-Excitation ResNet 50 (SEResNet50): We used the 3D Squeeze-and-Excitation ResNet SEResNet50 [82] from Monai [Medical Open Network for Artificial Intelligence MONAI https://monai.io/; last accessed 15 January 2022].

Residual neural networks (short: ResNets) [83] use “skip connections” to improve the vanishing gradient problem. In Wightman, Ross et al. [84], they are described as the “gold-standard architecture” for image classification tasks. Squeeze-and-Excitation Networks [82] add so-called “Squeeze-and-Excitation blocks” to existing convolution neural networks like the ResNets to improve channel interdependencies and outperform standard ResNets in image classification tasks. It was found that Squeeze and Excitation (SE) blocks combined with ResNet architecture performed outstandingly well in classifying brain tumours [83].

Own 3D CNN: Our developed network consists of a 3D CNN with two 3D convolutional layers as a backbone with two fully connected downstream layers. The first layer consists of one input channel for the 3D image, eight output channels (kernel size 3 × 3, stride 1) and a Relu activation function. The second layer has 16 output channels (kernel size 3 × 3, stride 1, 3D Dropout) and a Relu activation. The first fully connected layer reduces the feature size to 60 and the second one to the two output classes, “complete remission of disease” and “relapse of disease”. The output is passed through a softmax activation function.

## 3. Results

### 3.1. Patient Characteristics

A total of 30 consecutive patients with biopsy-proven MCL (4 women and 26 men; mean age 62.2 ± 9.7 years, range 42–76) met the criteria for participation in the study. An average of 4.4 lymph nodes per MCL patient (range 3–6) were analysed.

In total, 134 target lesions were evaluated. In this cohort, 17 patients (57%) were responders and in complete remission. 13 patients (43%) responded to first-line treatment but relapsed within the five year observation period.

All patients’ baseline clinical characteristics are summarised in Table 1.

### 3.2. Radiomic Analysis and the Machine Learning Prediction Model

Several traditional measurements of model performance were used, including sensitivity, precision, F1-Score, accuracy, ROC curve and the area under the ROC curve (AUC), to assess and compare the ML models. We used a principal component analysis (PCA) for dimensionality reduction while maintaining most of the variation in data to improve the accuracy of the predictive model. We used the grid search optimisation algorithm for fine-tuning and selected five components in our PCA.

Accuracy measures the model’s robustness and is defined by the percentage of correctly identified labels out of the entire population. Precision—the positive predictive value—is the probability that a predicted true label is indeed true. Sensitivity—the true positive rate (TPR) or recall—is the percentage of correctly identified true class labels. The F1-score is the harmonic mean of the sensitivity and precision.

The ROC curve is computed by plotting the recall versus the false positive rate at different decision thresholds. The AUC constructed from the ROC is the ability of the model to separate classes, with the number 1 standing for perfect separation and the number 0.5 standing for random classification. All tests were two-sided; *p* < 0.05 was considered statistically significant.

The predictive performances of the radiomics-based ML models to predict relapse of MCL disease are summarised in Table 2. The different models’ overall accuracy of predicting relapse was 61% (range: 58–64%, AUC = 0.58–0.62). The PCA KNN showed the best prediction accuracy.

Figure 5 shows the receiver operating characteristic (ROC) curves for the mean over the five runs. PCA KNN and KNN outperform the RF algorithm.

We performed a decision curve analysis to assess the clinical value of the predictive models in terms of net benefit against threshold probability. A decision curve analysis graph with threshold probability on the x-axis and net benefit on the y-axis illustrates the trade-offs between true positives and false positives, representing benefit and harm, respectively, as the threshold probability varies [13].

Figure 6 shows the decision curve analysis for the best run of the best performing machine learning model using PCA KNN.

### 3.3. The CNN-Based Neural Network Approach as Prediction Model

A total of 134 cubes were included for training. The results from five runs with five different seeds for the training and testing split were used for further comparative analysis. We evaluated the five scores’ accuracy, precision, recall, F1-Score and the area under the curve (AUC) by calculating the mean and standard deviation over the five runs.

The DL models’ overall accuracy of predicting relapse was 64% (range: 59–70%, AUC = 0.58–0.70). Based on the ROC analysis, our own 3D CNN significantly outperformed the 3D DenseNet and the 3D SEResNet50. Accuracy, Precision, Recall, F1-Score and AUC for the Own 3D Network is around 0.7 for the classification task.

The performances of the deep CNN models to predict the relapse of MCL disease are summarised in Table 3.

Figure 7 shows the receiver operating characteristics (ROC) curve for the mean over the five runs. For a False-Positive-Rate below 0.1 (= very high specificity), the 3D ResNet outperforms the Own 3D Network. For a False-Positive-Rate above 0.1, the own 3D network equals or mostly outperforms the 3D ResNet.

Analogous to the machine learning approach, we performed a decision curve analysis to assess the clinical value of the predictive models. Figure 8 illustrates the trade-offs between true positives and false positives in the decision curve analysis for the best run of the best performing deep learning model using our own 3D CNN.

### 3.4. Comparing the Deep Neural Network with the Machine Learning Approach

Figure 9 illustrates the metrics grouped by each data set and presents them as grouped bar charts for clarity and ranking of the different deep neural networks and machine learning models. 

Our own 3D CNN provided the best performance (AUC, accuracy, precision, sensitivity and F1-score were 0.70 ± 0.04, 0.70 ± 0.02, 0.71 ± 0.02, 0.70 ± 0.02 and 0.69 ± 0.01, respectively) among all deep and machine learning prediction models.

The deep learning model 3D SEResNet50 (AUC, accuracy, precision, sensitivity and F1-score were 0.62 ± 0.06, 0.62 ± 0.04, 0.65 ± 0.07, 0.62 ± 0.05, 0.60 ± 0.04, respectively) and the 3D DenseNet model performed worse in the comparison (AUC, accuracy, precision, sensitivity and F1-score were 0.58 ± 0.13, 0.59 ± 0.05, 0.64 ± 0.07, 0.59 ± 0.05 and 0.57 ± 0.06, respectively).

The machine learning model using KNN/PCA KNN showed the second-best result of all models (AUC, accuracy, precision, sensitivity and F1-score were 0.62 ± 0.02, 0.63 ± 0.02/0.64 ± 0.02, 0.64 ± 0.01, 0.63 ± 0.02 and 0.62 ± 0.02, respectively).

The PCA RF/RF-based ML models scored the worst (AUC, accuracy, precision, sensitivity and F1-score were 0.58 ± 0.02, 0.61 ± 0.04, 0.58 ± 0.02, 0.58 ± 0.02 and 0.58 ± 0.02, and 0.49 ± 0.07, 0.47 ± 0.07, 0.50 ± 0.09, 0.47 ± 0.07 and 0.45 ± 0.08, respectively).

## 4. Discussion

Mantle cell lymphoma is an aggressive disease associated with early relapse, chemo-refractory disease and poor long-term survival in most patients [13]. Standard of care emphasises aggressive treatment approaches, demonstrating the most prolonged durable remissions [7]. However, MCL has increasingly been considered a highly heterogeneous disease with subtypes with favourable disease features in which observation can be considered [1,85,86]. MCL is a “genomically unstable” malignancy associated with (sub)clonal heterogeneity and modulation of the genetic profile over time [15]. No routine biomarkers are currently established to predict short-term clinical outcomes. Radiomics refers to a fast-growing research area that converts standard-of-care imaging into minable high-dimensional data and builds subsequent predictive models to stratify patients into different risk groups based on the risk of occurrence of clinical endpoints, such as relapse of disease and personalise treatment [11,12,87,88].

As radiomic data convey information about tumour biology, such as temporal and spatial heterogeneity, they potentially reflect tumour behaviour and aggressiveness [26,88,89,90,91]. As AI techniques can analyse the tumour lesion as a whole and can be applied at multiple sites without the risk of a sampling error and biopsy-related complications, there is great potential for being used as a “virtual biopsy”.

One of the challenges in the classic machine learning algorithms relies on the several preprocessing steps, such as tumour segmentation, which regularly requires manual correction of the tumour boundaries computed by the (semi)automated algorithms which increases cost, time and the risk of inter-observer variation [92]. Deep learning can automatically perform these steps on the raw data and is considered a powerful analytical tool for different predictive data mining applications, especially in complex processes like biological systems [93,94,95,96]. Machine and deep learning-based artificial intelligence (AI) imaging techniques have the potential to support the decision making in clinical oncology for precise imaging. Of great clinical interest is the potential additive value of radiomics at baseline staging for early identification of high-risk patients to select the optimal therapeutic management and improve prognosis initially using the standard of care test, a simple contrast-enhanced CT. Moreover, radiomics holds the promising potential for disease monitoring, offering information related to disease evolution.

However, the current black-box approach makes it less acceptable for clinical application, particularly deep learning. To enhance the translation of radiomics into clinical practice, the reproducibility and stability of the features against technical perturbation are essential elements [97]. Unfortunately, there is often not one best-performing model from a statistical standpoint. Different classifiers can lead to models performing only slightly worse than or statistically similar to the best-performing model because radiomic features depend highly on acquisition parameters, hindering generalizability [98].

At best, the results should be validated using independent data sets, preferably using data from a different institution [26,99]. On a multi-institutional scale, a large amount of available medical imaging data exists, which constitutes a great opportunity for complex model training. But access for research purposes is highly restricted by law and ethics. And although the imaging biomarker standardisation initiative and quantitative imaging biomarkers alliance for standardisation give recommendations including postprocedural harmonisation and standardisation of the acquisition protocol for improved reproducibility [38,100], these methods are difficult to implement, particularly across multiple sites and concerning the constant innovations in medical imaging techniques [38,101,102]. Federated or distributed learning might be a potential solution in the future to perform local model development and training and to share the weights for collaboration [103]. According to the literature, internal validation techniques like ours can be used in small-scale pilot studies [104].

Most radiomics related studies focus on lymphoma classification, are PET/CT based and centre around Hodgkin lymphoma or diffuse large B cell lymphoma (DLBCL) [41,42,43,44,45,47,48,49,53,105]. However, there is no standard way to extract radiomics features, meaning reproducibility is a key challenge in this field [106]. 

Deep learning algorithms can detect or classify lymphoma; in line with radiomics studies, most rely on PET/CT imaging: Weisman et al. used 3D neural networks to automate lymph node detection and lymphoma assessment in PET/CT [107]. Li et al. used PET/CT for lymphoma characterisation [79] and Sibille et al. used DL to classify positive FDG uptake regions on PET/CT images as lymphoma or lung cancer [81]. Several studies address automated segmentation of lymphoma, especially in diffuse large B-cell lymphoma (DLBCL), which typically consists of larger lesion sizes and more intense signals on PET images than MCL [82,83]. Prognostication modelling could potentially be exploited to determine new and appropriate personalised medicine treatments [12]. However, only a few studies compared radiomics-based versus deep learning methods, for example, Bibault et al. regarding outcome prediction of rectal cancer [84].

There is little data regarding (AI)–based image analysis on MCL, which all used PET/CT imaging. Mayerhoefer et al. showed that PET/CT-derived radiomic features combined with clinical parameters lead to improved PFS prognostication [29]. Mayerhoefer et al. also deployed a machine learning approach for detecting bone marrow involvement of MCL on pelvic PET/CT images [108]. Zhou et al. used CNN architecture [109] to segment mantle cell lymphoma (MCL) in PET/CT images.

Also, the role of baseline semiquantitative 18F-FDG PET/CT parameters such as SUVmax for prognostication of MCL is not yet clear with controversial results [110,111,112,113,114]. Limitations are the heterogeneity between the study cohorts, such as therapy and baseline features. In the recent LyMa-PET Project, Bailly et al. demonstrated that SUVmax was the only independent prognostic factor for PFS and OS and suggested a scoring system combining MIPI and SUVmax for outcome prediction [112]. Albano et al. studied other baseline metabolic parameters like SUVbody surface area and demonstrated no correlation with PFS and OS, while for OS, only the MIPI score resulted as an independent prognostic factor in contrast to other common variables, including the Ki-67 score without a significant association with outcome survival [110]. So far, no strong evidence about the potential prognostic impact of 18F-FDG PET/CT on survival outcomes is available and more studies are needed to confirm or contradict these results.

To our knowledge, this is the first study that used two machine learning and three deep learning algorithms to investigate their potential as predictive models for early prediction of relapse of MCL. In this study, we compared the performance of RF and KNN—two traditional machine learning classifiers with three deep learning convolutional neural networks (3D Dense Net, 3D SEResNet50 and our own 3D CNN) to predict future relapse of MCL.

Our own 3D Net (AUC, accuracy, precision, sensitivity and F1-score were 0.70 ± 0.04, 0.70 ± 0.02, 0.71 ± 0.02, 0.70 ± 0.02 and 0.69 ± 0.01, respectively) provided the best performance among all deep and machine learning prediction models with a low variance in the five runs speaking for the stability of our model.

The sensitivity is a significant parameter for models’ evaluation as it reports how well the predictive model can detect MCL patients at high risk of relapse of disease.

The deep learning model 3D SEResNet50 (AUC, accuracy, precision, sensitivity and F1-score were 0.62 ± 0.06, 0.62 ± 0.04, 0.65 ± 0.07, 0.62 ± 0.05 and 0.60 ± 0.04, respectively) and the 3D DenseNet model performed worse in the comparison (AUC, accuracy, precision, sensitivity, and F1-score were 0.58 ± 0.13, 0.59 ± 0.05, 0.64 ± 0.07, 0.59 ± 0.05 and 0.57 ± 0.06, respectively). An explanation could be that the SEResNet50 and DenseNet have more learned variables, potentially leading to faster overfitting amid the input cubes of 20 × 80 × 70 pixels.

Interestingly, the machine learning model using KNN/PCA KNN showed the second-best result of all models (AUC, accuracy, precision, sensitivity and F1-score were 0.62 ± 0.02, 0.63 ± 0.02/0.64 ± 0.02, 0.64 ± 0.01, 0.63 ± 0.02 and 0.62 ± 0.02, respectively).

The PCA RF/RF-based ML models scored the worst (AUC, accuracy, precision, sensitivity and F1-score were 0.58 ± 0.02, 0.61 ± 0.04, 0.58 ± 0.02, 0.58 ± 0.02 and 0.58 ± 0.02, and 0.49 ± 0.07, 0.47 ± 0.07, 0.50 ± 0.09, 0.47 ± 0.07 and 0.45 ± 0.08, respectively) although random forests (RF) have displayed high predictive performance in several other biomedical applications.

The AUC results focus only on predictive accuracy and thus cannot prove a model is worth using. Metrics that concern accuracy does not incorporate information on consequences. Decision analysis incorporates consequences and thus can tell whether a model is worth using to identify the optimal prognostic model that maximises the outcome of interest [115].

In our case, a false-negative result—a patient being put in a low-risk group resulting in fewer follow-ups or missed maintenance therapy- is more harmful than a false-positive result resulting in closer follow-up. A model with much greater specificity but lower sensitivity than another would have a higher AUC yet yield a smaller net benefit and be a poorer choice for clinical use.

Prognostication modelling could potentially be exploited to determine new and appropriate personalised medicine treatments [12]. With the exponential increase in biomarkers and information technology advancements, prognostic and diagnostic models will likely increase. Reliable, comparable and easily understandable methods to determine the value of such models are required [116].

The number of studies on diagnostic and prognostic markers using accuracy metrics dwarfs those using decision analysis methods in the medical literature. Decision analytics is a great statistical tool without requiring further information, such as the costs or effectiveness of treatment. It can be a beneficial method directly applied to a model validation dataset.

The decision curve for our best-performing 3D CNN proved its value. It shows that between a threshold probability of 30% and 65%, checking patients based on our proposed classification model leads to a higher benefit than checking no or checking all lymphomas more frequently.

Although seemingly promising, this study is affected by several limitations that need further discussion. We note that the small sample size (*n* = 30 pts) did not allow for multiple testing corrections for the large number of radiomic features tested. However, statistical analyses were performed lesion-based (*n* = 134) as 3D-volumetric ROIs; To address the issue of the limited sample size, we applied recommended used techniques such as the bootstrap technique or data augmentation. Although our results showed sufficient statistical power, our proof-of-concept study remains hypothesis-generating, requiring further prospective validation as with other similar published studies before a wider adaptation into a clinical decision process.

Secondly, the nature of the study cohort: MCL is a less common lymphoma subtype than Hodgkin lymphoma or diffuse large B cell lymphoma (DLBCL) with smaller patient populations and more difficulties obtaining a sufficient number of cases with the event of interest. Furthermore, our cohort consisted of treatment-naïve MCL patients with in-house imaging and tissue confirmation at baseline and a documented five year follow up. MCL patients with progression were excluded for getting a distinct signature of relapse or remission. Including MCL patients with disease progression or a reference group with benign lymphadenopathy could be part of further analyses. In addition, this study was retrospective, raising potential patient selection biases as treatment was not randomly assigned to participants. Keeping this in mind, all extracted study data were randomly split into training and testing sets but stratified to ensure consistent distribution and mimic the real-world distribution of remission and relapse of disease. Thirdly, study data were collected from a single centre without external validation. Radiomic features depend highly on acquisition parameters, hindering generalizability [98]. But the reproducibility and stability of the features against technical perturbation are a prerequisite for developing quantitative imaging biomarkers used in clinical routine. Although there exist recommendations by the imaging biomarker standardisation initiative and quantitative imaging biomarkers alliance for standardisation [38,100], including postprocedural harmonisation and standardisation of the acquisition protocol for improved reproducibility, these methods are difficult to implement, particularly across multiple sites and concerning the constant innovations in medical imaging techniques [38,101,102].

Developing a prognostic model is a key tool for individualised therapy of lymphoma patients. Our pilot study compared the most popular classic machine learning techniques with advanced deep learning methods. While our proposed deep learning 3D CNN prediction model requires further improvement, we consider our preliminary results as a promising methodological basis for more extensive studies, ideally prospective and multicentric, that involve comparing and combining AI-based imaging models with the established MIPI score for improved staging and prognostication at diagnosis and as real-time surveillance for improved clinical decision-making amid innovative treatment strategies to pave the way for precise imaging, and thus improving the standard of care for mantle cell lymphoma.

## 5. Conclusions

This study demonstrates the application of deep neural networks and radiomics-based machine learning models to predict relapse in MCL on baseline CT studies.

The results revealed that (1) the own 3D CNN showed the best predictive performance compared to all deep and machine learning models (2) the machine learning model using KNN/PCA KNN showed the second-best results, superior to the 3D SEResNet50 and the 3D DenseNet and (3) the decision curve analysis proved our optimised 3D CNN a valuable predictive model. Further application of these predictive models might supplement future approaches for refined MCL diagnosis and assist workflows for personalised cancer medicine.

## Figures and Tables

**Figure 1 cancers-14-02008-f001:**
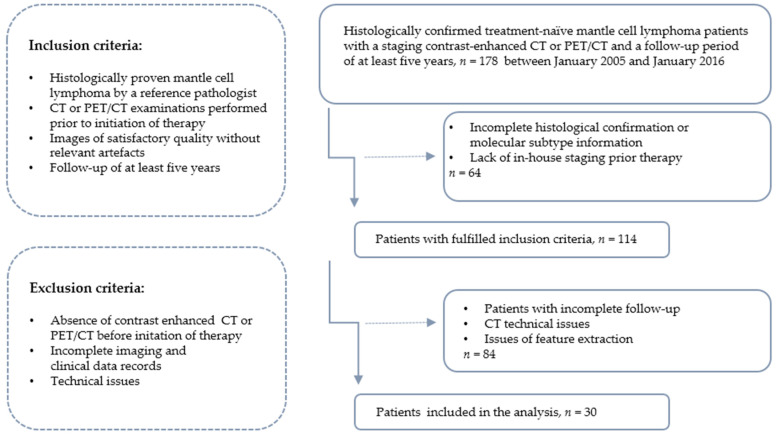
Recruitment pathway of the study.

**Figure 2 cancers-14-02008-f002:**
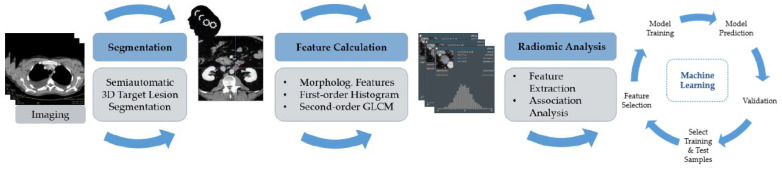
The schematic diagram for processing and analysis in the machine learning approach. Morphological features, including volume and textural features of first and second-order, were obtained. Details regarding the extraction settings are listed in Appendix A, Table A1.

**Figure 3 cancers-14-02008-f003:**
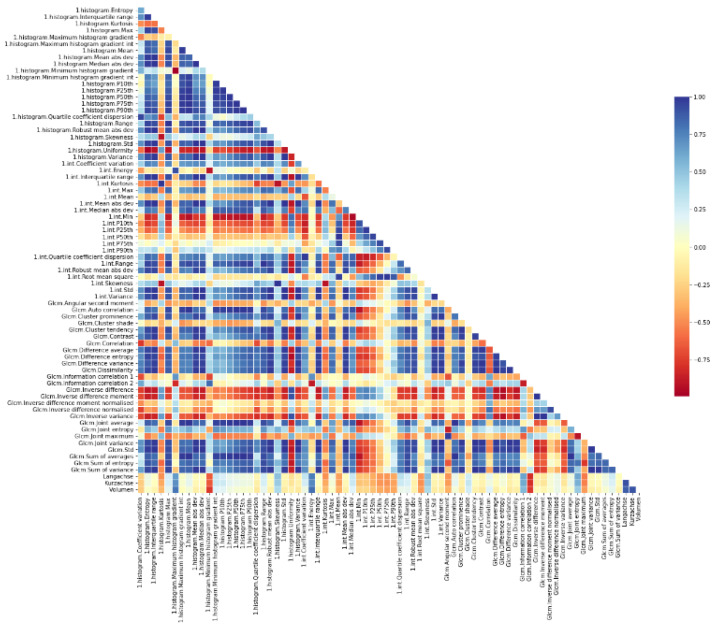
Correlation plot depicting the hierarchical clustering of all extracted features and their discriminatory power between the relapse and the remission group.

**Figure 4 cancers-14-02008-f004:**
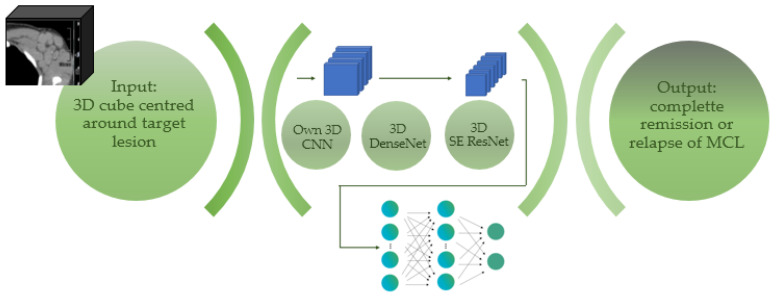
The schematic diagram for data processing in deep neural networks.

**Figure 5 cancers-14-02008-f005:**
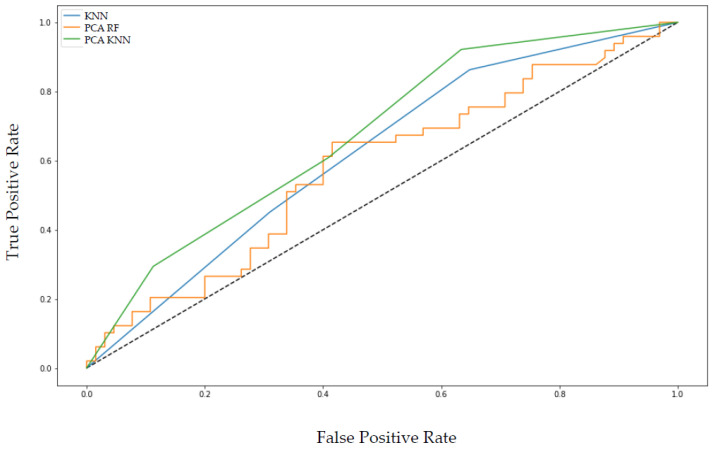
The receiver operating characteristics (ROC) curves of the KNN and PCA KNN algorithm yielded an area under the curve (AUC) of 0.62. In contrast, PCA RF yielded an AUC of 0.58 for classification performance of complete remission versus relapse of MCL.

**Figure 6 cancers-14-02008-f006:**
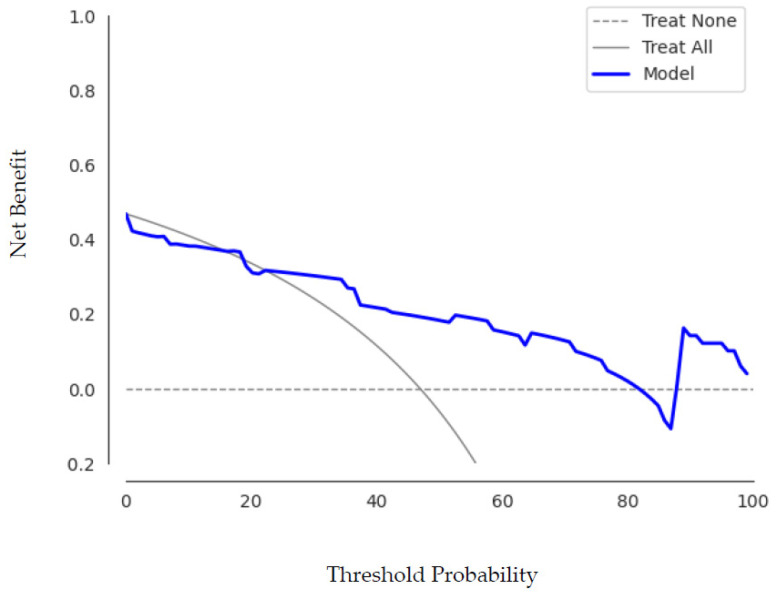
The decision curve analysis for the best run of PCA KNN shows that within a threshold probability from 25% to 55%, checking patients based on the classification model leads to a higher benefit than checking no or checking all lymphomas more frequently.

**Figure 7 cancers-14-02008-f007:**
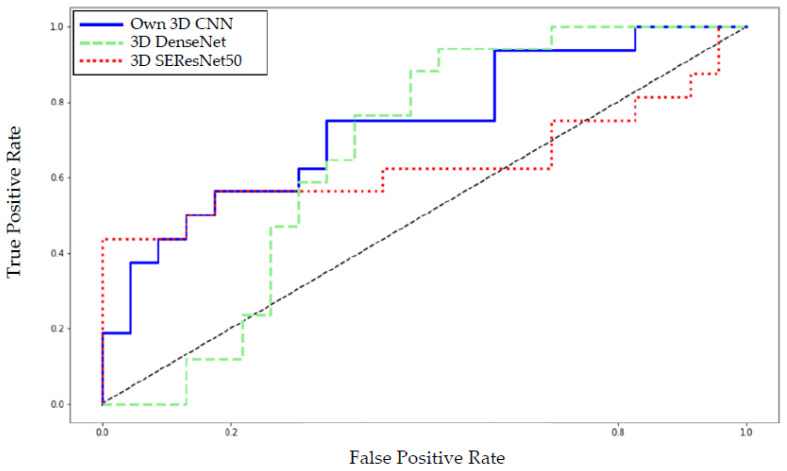
The deep learning models’ receiver operating characteristics (ROC) curve shows our 3D network mostly outperforming the 3D DenseNet and 3D SEResNet50.

**Figure 8 cancers-14-02008-f008:**
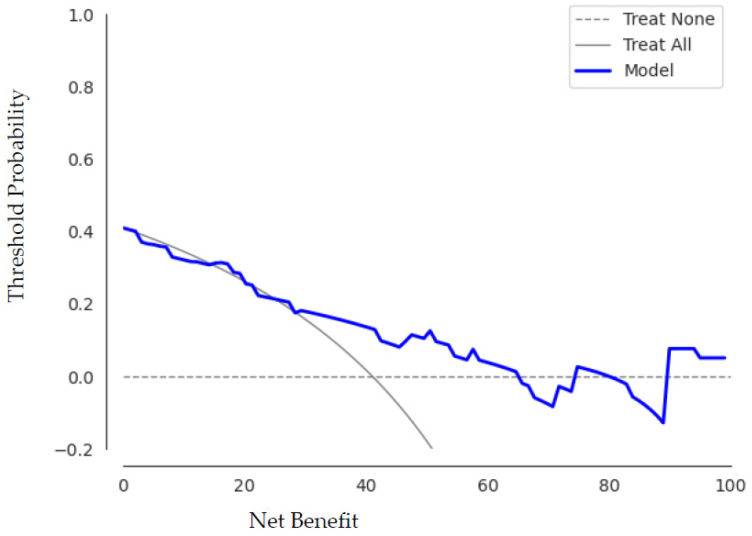
The decision curve analysis for the best run of the novel 3D CNN shows that within the threshold probability from 30% to 65% checking patients based on the classification model leads to a higher benefit than checking no or checking all lymphomas more frequently.

**Figure 9 cancers-14-02008-f009:**
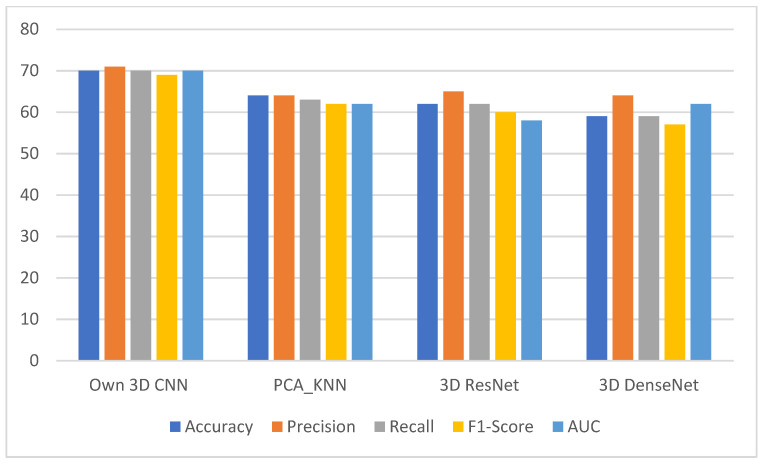
The grouped bar chart shows that our own 3D CNN has the highest performance among all prediction models.

**Table 1 cancers-14-02008-t001:** Baseline demographic and clinical data.

Characteristic	Number
Sex	
Male	26 (86.7%)
Female	4 (13.3%)
Average age (range)	62.2 ± 9.7 years (42–76)
Ann Arbor Stage	
Stage I	0
Stage II	2 (6.7%)
Stage III	5 (16.7%)
Stage IV	23 (76.7%)
Patients’ status in 5-years follow up	
Complete remission (CR)	17 (57%)
Relapse of disease (RD)	13 (43%)

**Table 2 cancers-14-02008-t002:** The outcome of the radiomics-based machine learning models.

Machine Learning Models	Accuracy	Precision	Sensitivity	F1-Score	AUC
KNN	0.63 ± 0.02	0.64 ± 0.01	0.63 ± 0.02	0.62 ± 0.02	0.62 ± 0.02
PCA KNN	0.64 ± 0.02	0.64 ± 0.01	0.63 ± 0.02	0.62 ± 0.02	0.62 ± 0.02
RF	0.47 ± 0.07	0.50 ± 0.09	0.47 ± 0.07	0.45 ± 0.08	0.49 ± 0.07
PCA RF	0.58 ± 0.02	0.61 ± 0.04	0.58 ± 0.02	0.58 ± 0.02	0.58 ± 0.02

**Table 3 cancers-14-02008-t003:** The outcome of the deep learning models.

Deep Learning Models	Accuracy	Precision	Sensitivity	F1-Score	AUC
Own 3D Net	0.70 ± 0.02	0.71 ± 0.02	0.70 ± 0.02	0.69 ± 0.01	0.70 ± 0.04
3D DenseNet	0.59 ± 0.05	0.64 ± 0.07	0.59 ± 0.05	0.57 ± 0.06	0.58 ± 0.13
SEResNet50	0.62 ± 0.04	0.65 ± 0.07	0.62 ± 0.05	0.60 ± 0.04	0.62 ± 0.06

## Data Availability

The data presented in this study are available on request from the corresponding author. The data are not publicly available due to privacy restrictions.

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
