# Peer review of "Deep Neural Networks and Machine Learning Radiomics Modelling for Prediction of Relapse in Mantle Cell Lymphoma"

_cancers, 2022, doi:10.3390/cancers14082008_

Round 1
Reviewer 1 Report
Interesting methodological approach. However, the limits of the study as admitted by the Authors is the retrospective study design and the small sample size. This last point is crucial for supporting the prognostic impact of this analysis. Moreover, also for a clinician expert of MCL is difficult to understand how a sophisticated method of lymph nodes analysis could help to predict the clinical outcome of MCL patients. In spite of the comparison of different methodological methods this point should be better and convincingly explained in the paper in order to improve its value.
Reviewer 2 Report
The paper proposed by the authors is very interesting and original. The potential impact of radiomics in lymphoma setting is emerging, especially considering PET texture analysis. About CT radiomics analysis, no previous similar papers are available. So it is an absolute novelty!!
I suggest to accept the paper after some modifications:
- one of the major bias of the study is the sample, only 30 patients. It is true that 134 lesions were analyzed, but also 124 is not a so big number considering all the radiomics features available. This point needs to be underline better
- period of inclusion 2005-2018 is very long, did the patients used the same CT? the same PET? the same protocol? It seems very strange
- in the methods you wrote that at least 5 years of follow-up was available for each patient, but inclusion time period is until 2018....
- the introduction is well written but too long. Can you reduce it?
- it is not clear what features do you include. histogram? homogenenity? Can you put a supplemental table with all features included in your models
- The results of this work are amazing, however it lacks something about MCL-IPI, clinical features. Are your models better than clinical features? a combination? can you discuss this point at the discussion at least.
- a big forgetfulness is present in the paper. PET/CT and its role in treatment response and prognosis is already demonstrated in literature. It is crucial to write few lines about this topic.
Round 2
Reviewer 2 Report
After modifications, the paper is clear and complete